# Policy Development on Upskilling/Reskilling Older Population Care Staff in China

**DOI:** 10.3390/ijerph19159440

**Published:** 2022-08-01

**Authors:** Jason Hung

**Affiliations:** Department of Sociology, The University of Cambridge, Cambridge CB3 0SZ, UK; ysh26@cam.ac.uk; Tel.: +44-07478119080

**Keywords:** elderly care, healthcare, older adults, policy development, China

## Abstract

Mainland China has been concerned about the national growth rate of older adults aged 60 or above. The rapid growth of the cohort of older adults will significantly burden the Chinese healthcare system as they are at higher risk of suffering from chronic illnesses and functional disabilities. In geriatrics, aged populations often endure a wide range of diseases, dysfunctions, and cognitive impairment, so the corresponding healthcare services needed for them are substantial. The rise in the older adults’ life expectancy has compounded the burden of the healthcare system in mainland China in the long term. In this narrative essay, it is important to discuss how the state should assume a higher share of relevant responsibilities, by assessing how Chinese policymaking has been transformed to better satisfy the older population’s care and healthcare needs in mainland China. It is also pivotal to focus on analysing relevant Chinese policy development within the most recent dozen years to address how China’s state and local governments have been progressing in promptly providing health and older population care services to older Chinese adults. Because of the supply shortage and low quality of older population caregivers and alternative professionals, it is necessary to discuss and highlight the need to reskill or upskill relevant caregivers. As the trend of rural-to-urban labour migration continues, working adults of rural origins increasingly cannot provide domestic older population care, and human investment in training caregivers is an urgent task of Chinese policymaking. Therefore, how Chinese policymaking encourages upskilling or reskilling relevant caregivers is examined in this narrative essay.

## 1. Introduction

Mainland China has been concerned about the national growth rate of older adults aged 60 or above. The annual growth rate of older adults was 3.37% in the 1990s, a figure nearly three times the country’s overall population growth rate [1]. As per the China Research Centre on Ageing, there were 202 million older individuals in China in 2013, with 23 million aged 80 or above [2]. Among the older adults in 2013, 100 million and 37 million suffered from non-communicable diseases and disabilities, respectively [2]. Providing accessible, affordable long-term care (LTC) services to older adults has, therefore, become an urgent task for the Chinese Government [3]. While the provision of LTC services to older adults remains at a pre-mature, regionally piloting stage in the mainland, neighbouring countries/regions and advanced Western nations have already developed more efficient, sustainable responses to the ageing concern. For example, Japan, South Korea and even Taiwan have built structured LTC insurance systems [4]. Japan, the United States and Western Europe have additionally reskilled caregivers and other professionals to use social robots for older population care services [5]. As the life span continues to grow across all States, ageing will become, presumably, a more universal problem. As mainland China has been endeavouring to strengthen its position as a superpower, it is necessary to discuss how the Chinese Government responds to the problem of older population care, given the fact that its, neighbouring or not, advanced counterparts have already constructed relatively better-designed and more mature and sustainable older population care systems.

The ageing population is projected to reach 240 million in 2030 (accounting for 16% of the national population) and 450 million in 2050 (accounting for 33% of the national population) in mainland China [3]. Such forecasts have alarmed Chinese policymakers, as mainland China will turn from an ageing population to an aged population. An increase in life expectancy and a decline in fertility rates have been attributed to the country’s socio-demographic transformation. The rapid growth of the cohort of older adults will burden the Chinese healthcare system as they are at higher risk of suffering from chronic illnesses and functional disabilities [6]. In geriatrics, aged populations often endure a wide range of diseases, dysfunctions and cognitive impairment, so the corresponding requirement for healthcare services is substantial [7]. The rise in older adults’ life expectancy has compounded long-term burdens on the Chinese healthcare system [8].

The socio-economic transformation, the growing number of rural-to-urban migrants and the low fertility rates propel significant difficulties for urban, and especially rural, Chinese older adults to demand family care. The availability of nursing staff, particularly in rural China, is limited, and the current supply of urban, and especially rural, healthcare services can hardly support the rising demand for older population care [9]. Without a sufficiently inclusive older population care system, more Chinese cohorts are encountering poverty because of their lack of health insurance to ensure sufficient coverage of medical expenses for the older population [3]. Unlike urban older adults, their rural counterparts are insufficiently supported by pensions, so medical expenses become a significant financial burden for their families [1,10]. Moreover, very few geriatric hospitals and nursing homes are available in villages, towns and cities to support the medication, rehabilitation and care services for older adults [2].

Traditionally, Chinese families emphasise the importance of filial piety, where families assume the main responsibilities of care for older adults, as per the values of Confucianism [1,9]. However, the modernisation of mainland China, the decline of the size of working-age populations and the limited availability of family care due to rural-to-urban labour migration all challenge the provision of sufficient family care for older adults [11,12]. For example, some 50 and 37% of the older adults in cities and villages, respectively, live in empty-nest families [3]. An empty nest refers to families where the grown-up children move out for study, work, marriage or alternative reasons, leaving the older adults behind [8,11]. The growth of the number of empty nesters has diminished Chinese families’ ability to take care of their aged dependents [13]. In this narrative essay, it is important to discuss how the state should assume a higher share of relevant responsibilities by assessing how Chinese policymaking has been transformed to better satisfy the older population care and healthcare needs in mainland China.

Local governments may formulate and deliver their own designed older population care services and any associated services charges within the country. The Ministry of Civil Affairs (MCA) may consider subsidising lower-income local governments to implement their policies if such local governments can expectedly meet the investment targets [3]. Here, local governments must design specific policy guidelines and funding arrangements to develop their LTC systems [3,4]. Local governments face financial barriers to delivering relevant services. This is primarily because direct funding given by the Central Government to older population care is limited. While the Chinese older adults who are entitled to sufficient financial and social assets can purchase needed older population care services from the market, their underprivileged counterparts fail to enjoy adequate services [9]. In this narrative, it is pivotal to focus on analysing relevant Chinese policy development within the most recent dozen years to address how the state and local governments of mainland China have been progressing in providing health and older population care services to Chinese older adults.

An increasing proportion of the elderly population, even those that are rural, believe that the traditional approach of hiring nannies to provide older population care services fails to satisfy the provision of high quality of life for the older adults themselves, so they prefer pursuing professional, formal older population care arrangements [14]. For example, those with “three Nos” (i.e., no children, no relatives, and no homes) may choose to live in nursing homes when their family members cannot provide older population care [15]. However, most nursing homes, including those run by private actors, are poorly equipped and cannot fully satisfy the physical and psychological needs of the elderly population [4,12]. An alternative old-age care establishment is a community-based older population care centre (CBECC). CBECCs differ slightly from nursing homes, as the elderly population living in the former, but not the latter, can freely decide if they want to spend time at home, such as during weekends [4]. Here, community care refers to mutual care, cultural bonding and collective responsibilities for the common good. It is a major social service delivery and welfare distribution model in mainland China [6,16].

Despite the availability of older population care services in mainland China, the quality and supply of nursing staff remain a concerning challenge. The prolonged working hours, low wages, heavy workloads and emotional burdens render high staff turnover rates [6]. Such caregivers receive little training and have limited, if any, qualifications [17]. The lack of relevant professional knowledge and skills to deliver caregiving bars them from providing satisfactory service quality [3,6,14]. As mainland China’s provision of professional older population care services is nascent, it is necessary to discuss and highlight the need to reskill or upskill the caregivers. As the trend of rural-to-urban labour migration continues, working adults of rural origins increasingly cannot provide domestic older population care, and ensuring investments in training caregivers has become urgent for Chinese policymaking [18]. Therefore, how Chinese policymaking encourages reskilling or upskilling relevant caregivers is examined in this narrative essay.

## 2. The Challenges of, or Works to, Upskilling/Reskilling Older Population Care Staff

Mainland China needs at least 10 million trained older population caregivers [3]. However, mainland China encounters substantial barriers to recruiting professionally trained staff, including nurses, personal care workers and social workers. The long-term care services provided in the country are usually insufficient and of poor quality [3,19]. This is partly because most older population care services staff, especially those working in private nursing homes, are rural-to-urban migrant workers or previously laid-off workers who receive no formal training in caregiving [19]. These unprofessionally trained caregivers also lack the relevant educational qualification to prepare themselves to undertake the roles in older population care [17]. Therefore, service staff training is urgently needed in mainland China, where staff should engage in a wide range of professional training programmes covering basic older population care skills, communication skills with older adults and ethical education [19]. When workers are laid off for a prolonged period, their human capital will depreciate, and they may experience difficulties returning to the labour market. Therefore, local governments should offer professional training to those former workers who left the labour market for a while; otherwise, their values in contributing to the caregiving industry may be limited [18]. Song et al. (2014) argue that almost all older population caregivers (i.e., 96%) are willing to participate in training that leads to more knowledge and skills [17]. Therefore, such an argument optimistically indicates that upskilling or reskilling the older population caregiving staff is realistically achievable, as long as relevant training programmes are offered.

However, older population caregiving institutions often lack the funding to upgrade their medical equipment and recruit highly qualified doctors and other medical staff [1]. While evaluating the quality of caregiving services provided by these institutions is required for monitoring purposes, local governments cannot operate a comprehensive service evaluation mechanism and a quality control system to ensure the services delivered are satisfactory [11]. The state government should allocate more financial resources to the local governments to hire better-skilled and more qualified older population caregiving staff, alongside building and running a quality control system within the older population caregiving institutions. Here, performance-based mechanisms can be implemented, meaning that decent service providers can be rewarded with more government subsidies to enhance, or at least maintain, the quality of the services [19].

Li et al. (2022) employed a convenience sampling approach to recruit 589 older individuals aged 65 or above from two geriatric hospitals and six nursing homes, 415 medical staff from one community institution, seven geriatric hospitals and five geriatric nursing homes and 372 nursing staff from 21 geriatric nursing homes [14]. These respondents were asked to complete an online survey between July and August 2020. The findings revealed that only 53.55 of medical and nursing staff obtained a work qualification certificate [14]. While employed, 83.6% of nursing staff engaged in relevant training, while 62.4% of staff received training at least once per month. The primary purpose of the training was to enhance professional skills (87.1%) and earn relevant practising certification (53.4%). Further findings showed that the major challenges with professional training were the lack of training opportunities (40.8%) and inappropriate duration or location of training (35.7%). This means, for example, training was delivered in inappropriate time slots [14]. Among the 83.6% of nursing staff who participated in on-the-job training, as little as 53.4% believed the training content was conducive to their work, and merely 38.9% were satisfied with the training quality [14]. These findings suggest that more practical training should be made available to all medical and nursing staff, so staff are more likely to attend training sessions in the time slots where they are not occupied by work. Moreover, the training contents should be reviewed and updated regularly to ensure those who attend the training sessions can learn practical knowledge and earn useful skills and certifications that are conducive to their future career in older population care.

Zeng et al. (2019) conducted another survey between April and June 2016 in Zhejiang Province of China [6]. A multi-stage, stratified cluster random sampling approach was employed to survey 84 caregivers. Their findings revealed that pre-employment training positively influenced nursing knowledge and attitudes. On-the-job training also positively influenced nursing practices [6]. These findings echo results from Li et al. (2022) and suggest adjustments in Chinese policymaking where more professional training should be made available to caregivers.

## 3. Chinese Policy Development on Upskilling/Reskilling Older Population Care Staff

Diagram 1 summarises and presents Chinese policy development on human investment in upskilling or reskilling older population caregivers and alternative professionals within the industry. Interestingly, the MCA implemented the National Medium and Long-term Development Plan for Civil Affairs Talents (2010–20) in 2011. In 2014, nine ministries, including the Ministry of Education and MCA, collectively delivered the Opinions of the State Council on Accelerating the Training of Talents in the Elderly Care Service Industry. In 2020, the Ministry of Human Resources and Social Security, the MCA, the Ministry of Finance, the Ministry of Commerce and the All-China Women’s Federation Circular jointly implemented the Health Care Vocational Skills Training Plan. From a broad perspective, in the 2010s, mainland China’s policy development on human investment in the older population care industry had turned from single ministerial/departmental to multiple ministerial/departmental endeavours. Such a circumstance shows that the state government had increasingly emphasised upskilling/reskilling older population care staff.

In the recent two national Five-Year Plans (FYPs), the state government has endeavoured to prioritise the arrangement of the older population and health care services to those in need. The 13th FYP (2016–2020) highlighted that medical and nursing services with traditional Chinese medicine characteristics would be developed. The change included constructing new nursing homes focusing on traditional Chinese medicine for healthy older population care. In addition, clinics focusing on the prevention and treatment of geriatric diseases and chronic diseases would be established. Such policies reflect that the state government stressed the importance of incorporating healthcare into older population care services. Furthermore, the 13th FYP asserted that family doctors’ services would be actively promoted, teams of social care workers would be cultivated, long-term care services for the elderly at home would be provided and care training for family members would be provided. Here, the state government combined multiple parties to collectively deliver older population care and healthcare services to the aged populations. Furthermore, the 13th FYP stated that grassroots medical and health institutions would be built to actively carry out services such as medical treatment, rehabilitation and nursing, and improve the accessibility of medical and health services for older adults. Moreover, free-of-charge clinics for older adults would be in service, and door-to-door services for older adults with mobility difficulties would be provided. Such a policy ensured that older adults of, financially or otherwise, less-privileged profiles could also benefit from basic older population care and healthcare services.

The 14th FYP (2021–2025) highlights relevant human investment policies in the older population care industry. For example, the state government emphasises organising and coordinating volunteers to provide care services for disabled older adults at home and encouraging social forces to use community supporting housing to set up nursing stations to provide home health services for the disabled elderly. Where experimental pilot schemes had been used, they also introduced palliative care services for older adults. Here, the state government aims to improve the multidisciplinary service model of palliative care and raise the quality of life of terminally ill patients (see Figure 1). In addition, the 14th FYP introduces a new health promotion concept for older adults. Here, the older adults are guided to take “maintaining body functions and the ability to live independently” as their health goal; and establish the awareness of “their own health is primarily their own responsibility (see Figure 1).” Such a concept strengthens the idea that “the family is the first gateway to health,” and they encourage the older adults to practise a healthy lifestyle alongside their families.

To ensure more new blood enters the older generation care and healthcare industry, the 14th FYP assured that the state government supports education institutions to incorporate older generation health education into the curriculum content. They prompt colleges and medical and health institutions offering medical majors to set up opportunities for older generation health education and offer courses such as health care and nursing skills training to the aged populations, their families and caregivers. The 14th FYP details show that the state government has been emphasising the requirements for skillsets acquired by older generation care and healthcare professionals.

## 4. Conclusions

Low wages, low socio-occupational status and an inadequate number of individuals interested in working in older population care are deemed barriers to providing a steady supply of human resources. The state government should exercise better arrangement and delivery of financing to ensure human resources, especially in poorer rural regions, are made available. Here, the Chinese government needs to develop mechanisms and long-term solutions to raise the monetary and non-monetary incentives (such as professional certificates) for the purpose of retaining older population care professionals and caregivers in less appealing, rural regions for work. Otherwise, the turnover rates of rural caregivers and professionals within the older population care industry would remain high, resulting in a significant shortage of the supply of nursing staff, social workers and other specialists to take care of rural older adults.

It is noteworthy that the 13th FYP and 14th FYP have emphasised a focus on recruiting or training caregivers to diversified specialists within the older population care sector. These specialists include doctors and nurses in geriatric hospitals and Information and Communications Technology (ICT) talents. Traditionally, the low-skill requirement and low-socio-occupational status in older population care have barred new blood from entering the industry. However, to date, mainland China has been increasingly diversifying the occupational positions in older population care, and more high-skilled labours and talents are needed to support the industry. Therefore, in the long-term, the socio-occupational stigmatisation against labour in older population care should be minimised, as more better-skilled and -paid professionals have been entering the market. This expectation is supported by relevant policy development in mainland China where the state government has been recruiting more young labourers possessing a range of specialised skillsets for entry into older population care. Additionally, untrained or less-trained caregivers are encouraged to attend vocational training programmes where professional certificates are granted upon the completion of the training programmes. As a result, more existing caregivers in the industry have been upskilled, allowing them to assume more specialised occupational responsibilities and secure higher wages or salaries in the long term.

Education is pivotal to improving the capacity building of Chinese policymakers for human resources development at both the local- and national levels. For example, workshops should be arranged and organised to develop relevant Chinese policymakers’ understanding of the educational sector’s challenges, to help them better and more accurately respond to redesign the educational curriculum for training older generation care and healthcare professionals in mainland China. Moreover, workshops and seminars should be delivered to ensure Chinese policymakers from each ministry or department recognise their roles and responsibilities, alongside understanding how they can develop joint-ministerial policy development activities or operations when upskilling/reskilling older population care staff. Otherwise, miscommunications and confusion between ministries and the duplication of works would plausibly arise, discouraging the efficiency of relevant Chinese policy development.

## 5. Additional Thoughts to Enrich Relevant Scholarly Debates

Human resources and innovative technological development are increasingly intertwined in Chinese policymaking. As early as the implementation of the 12th FYP (2011–2015), the state government already placed a priority on the development of home-based community care services, including basic at-home nursing and rehabilitation care services [7]. Such policy development paved the way for mainland China to develop a smart and healthy older population care industry, where local governments have been reoriented to build smart older population care facilities at institutions and residential units. Here, smart older population care means the use of technologies to collect, store and analyse big data in order to arrange older population care services accordingly [20]. The 12th FYP indicated that home-based older population care services in mainland China are quasi-public goods, promoting both local governments and commercial companies to individually, but also collaboratively, provide innovative technological services [20].

The incorporation of innovative technologies in older population care facilitates the convenience, efficiency and quality of caregiving. Therefore, in 2014, for example, the MCA implemented a national-level project for the employment of smart homes in aged care in seven nursing institutions. Here, technologies have been utilised to conduct physical examination, sleep monitoring, falling detection and other healthcare services [9]. However, to date, in mainland China as a whole, innovative technological development in older population care is still premature, in part caused by the lack of relevant human investment within the industry. For example, older adults themselves, especially those of rural origins, are reluctant to accept new, innovative ideas; government authorities lack the necessary capacity to develop guidelines on how to concretely market smart older population care development. Moreover, government authorities fail to design the vocational educational curriculum for the purpose of allowing caregivers and other specialists to learn how smart home devices can be aptly used. Personalised behaviour data of the older adults are also insufficient when building user-friendly smart home older population care facilities [21]. Therefore, from the standpoints of customers, older population care professionals and government authorities, human resources development should be emphasised in order to optimise the outcomes of creating and using innovative technologies in older population care.

It is necessary to acknowledge that older adults suffering from physical and cognitive impairments may fail to use technological, smart devices for better self-care. Here, their family members or close friends, alongside the caregivers and healthcare professionals, should be upskilled/reskilled in order to help execute smart functions for the impaired older populations. To date, the design of smart devices for older population care in mainland China is rather simple. Therefore, functionally abled older adults, custodial workers and family members of older cohorts, to a large extent, only need to develop basic skills to use, and familiarity with using, innovative technological products. However, nurses, general practitioners and especially specialist doctors should be expected to complete relevant, more advanced on-the-job training and a long duration of medical education in order to incorporate innovative technological tools into their skilled professional practices. Here accredited certification should be provided upon completion of on-the-job training and education, for the purpose of improving the credibility of older population caregivers and healthcare professionals.

The Asian Development Bank, for example, has been partnering with local Chinese governments at the regional level to design (vocational) education curricula to teach older population caregivers and specialists how to use innovative technologies to take care of older cohorts. The expanded scope of local Chinese governments should endeavour to design or amend their (vocational) education curriculum in order to ensure lower-skilled caregivers and better-skilled specialists can all learn the relevant, up-to-date educational outputs and practise more innovative caregiving and/or healthcare services. More vocational colleges and higher education institutions should arrange and deliver older population caregiving training and educational programmes to support the education curriculum designed by local governments. Costs involved should be covered, or at least subsidised, by the State Government of China or multinational corporations such as Asian Development Bank.

It is observable that the Chinese Government has adopted a multi-faceted approach when designing, arranging, delivering and evaluating older population caregiving and related healthcare services. For example, pilot studies were employed in Shanghai and Luoning County in the 2010s to test the efficiency and effectiveness of upskilling/reskilling personnel and setting up smart services [9]. Concurrently, the FYPs have been issued to highlight, policy-wise, how mainland China, at large, should engage for the purpose of strengthening the sustainability of older adult caregiving and healthcare services. Therefore, both higher- and lower-level governments play roles in either developing general policies on older population care and related healthcare or executing such policies regionally. Yet, under the restrictions of bureaucracy, lower-level governments may receive confusing or conflicting orders from their upper-level governments. For example, lower-level governments may be ordered by multiple upper-level governments to implement different older age population care policies. Better vertical communication must be arranged in order for mainland China to efficiently build a more sustainable ageing society [22].

Additionally, to better discuss upskilling/reskilling older population care staff in mainland China, scholars and policymakers can focus on the discourse on how innovative training methods can be used to upskill older population caregivers and nurses. furthermore, they can particularly educate older population caregivers on how to manage aged care in residential units, where smart home devices are installed. Government authorities should monitor the quality of the training and vocational education in order to ensure such human investments are arranged and delivered effectively and impactfully. Especially during the post-pandemic epoch, technological transformation in elderly care has been accelerated. If proper training and vocational training can be delivered to older population care professionals, a wider variety of services, such as long-distance consultations, can be employed to reduce the costs related to elderly care. As there is a shortage of professional, better-skilled medical staff in villages, long-distance consultations allow urban doctors to conduct virtual medical examinations on rural dwellers. Therefore, given the positive outcomes of having a mature, innovative, technological, older population care setting, human resource development should further be prioritised to educate multiple parties on how to maximise their skillsets to serve beneficiaries of aged care in a cost-effective, convenient and efficient fashion.

## Figures and Tables

**Figure 1 ijerph-19-09440-f001:**
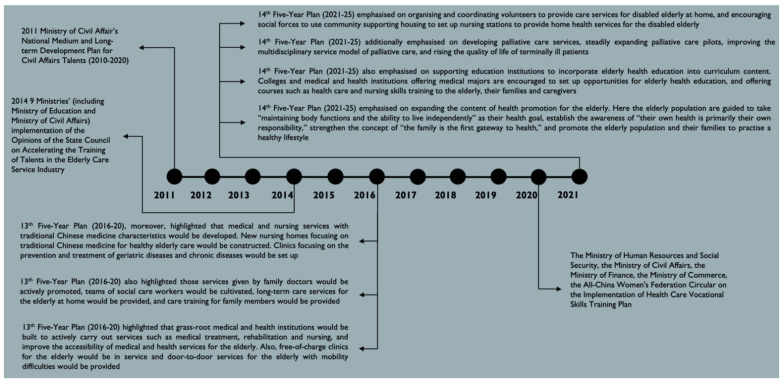
The Timeline of Older Population Care Policy Development on Human Resources Development in China.

## Data Availability

Not applicable.

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
