# Peer review of "Policy Development on Upskilling/Reskilling Older Population Care Staff in China"

_ijerph, 2022, doi:10.3390/ijerph19159440_

Round 1

Reviewer 1 Report

An interesting and very well narrative essay on the need to upskill/reskill staff who provide elderly care in China. The first point worth noting is that this appears to be a common phenomenon across affluent States in the international community. The need to provide appropriately skilled staff to care for the increasing numbers of elderly persons is a worldwide concern. This is not a problem particular to China. Or, indeed, is it? Accordingly, it would be good to incorporate some comparisons with other States or international data that may be available through the UN. As the 'life span' continues to increase across all States this will become, presumably, more and more a universal problem.

My second observation is that all health care systems appear to be searching for technological solutions for the provision of health care services. Health care for the elderly is no exception. This was most noticeable during the pandemic when social distancing was necessary. However, there are some unique barriers to apply technological solutions effectively for elderly care delivery, especially, as the person ages and may suffer physical and cognitive impairments. This should be acknowledged and considered in more detail, in my opinion.

The third point has to do with the degree of upskilling and reskilling required for elderly care staff and, particularly, for those who work with people in their 'at home settings,' as opposed to nursing homes, and, long-term care facilities, etc. The degree of upskilling and reskilling will depend on the functional role that the elderly health care staff will be expected to fulfill. This could range from specialist doctors to custodial workers. This does not seem to be covered in sufficient detail in this submission. Should there not be a clear delination in the types of elderly health care staff needed and the degree of training/education that they might be required to be eligible to fulfill their unique individual roles. Presumably, this also applies to elderly health care administrators.

There seems to be too little attention paid to elderly health care education and training requirements and, more specifically, accreditation. Presumably, this will be the responsibility of universities and colleges. Does this entail a further consideration of the existing curriculae at the Faculties of Medicine and Health at these institutions of higher education? Does it also require more research in elderly care diagnosis, treatment and delivery? If so, who will cover the cost of expanding the capacity of these institutions of higher learning to meet the needs of our growing elderly populations? 

China is a very diverse and vast country with something like 23 provinces, five autonomous regions, four municipalities, and two Special Administrative Regions. There is a reference to the rural-urban divide but no consideration of how this concern plays out across the different administrative units at the sub-national level in China. Have any of these administrative units made any progress on this concern or may have set the standard and pace for others to follow?

And, lastly, it is a well known fact that China has the highest population in the world with some 1.42 billion people. What particular challenges does this present to the country on this issue and concern? Is it easier to scale up the number of elderly health care skilled staff or make it harder to do so because of the size of the population?

On the other hand, given it has a single party Communist system of government, that is noted for its authoritarianism and repression of its population, it might be easier to adopt and to implement the types of elderly health care staff upskilling and reskilling needed to service its growing elderly population. 

I noticed, as well, a number of your endnotes are off. See lines 336, 346, that have noticeable gaps.

Author Response

  1. The author added a paragraph presenting how neighbouring or advanced Western countries/regions have already developed more sustainable, better structured Long-Term Care (LTC) services for older populations. When mainland China has been endeavouring to strengthen its position as a superpower, it is necessary to discuss how the Chinese Government responds to the problem of older population care as their current form of LTC services is still premature. 
  2. A paragraph was added to describe how impaired older adults may not be able to use smart devices, and how their family members, closed friends or custodial caregivers can serve as the alternative individuals to use smart services for older adults.
  3. A paragraph was added to explain how different significant parties should receive different levels of on-the-job training and vocational education for the delivery of healthcare services or basic caregiving.
  4. Content about the need to provide accredited certification was added.
  5. How upper-level governments focusing on arranging and delivering ageing policies and lower-level administrative units, such as Shanghai and Luoning County, piloting smart older population care services were mentioned.

Reviewer 2 Report

This is a study to address human resource development in elderly care analyzing policies.

The findings of this study could contribute to settings where health social welfare system for health of older adults is limited.

Authors may want to add a paragraph or to discuss skills and trainings that Chinese policies aiming for care givers at different levels and profession (nurse, social and care worker, community volunteers) in the section three.

Author Response

A paragraph to discuss skills and trainings that Chinese policies aiming for caregivers at different levels and profession was already added.